# Structural Diversity and Carbon Dioxide Sorption Selectivity of Zinc(II) Metal-Organic Frameworks Based on Bis(1,2,4-triazol-1-yl)methane and Terephthalic Acid

**DOI:** 10.3390/molecules27196481

**Published:** 2022-10-01

**Authors:** Taisiya S. Sukhikh, Evgeny Yu. Filatov, Alexey A. Ryadun, Konstantin A. Kovalenko, Andrei S. Potapov

**Affiliations:** Nikolaev Institute of Inorganic Chemistry, Siberian Branch of the Russian Academy of Sciences, 3 Lavrentiev Ave., 630090 Novosibirsk, Russia

**Keywords:** metal–organic frameworks, bis(1,2,4-triazol-1-yl)methane, terephthalic acid, coordination polymers, luminescence, gas adsorption, gas separation

## Abstract

A three-component reaction between the 1,4-benzenedicarboxylic (terephthalic) acid (H_2_bdc), bis(1,2,4-triazol-1-yl)methane (btrm) and zinc nitrate was studied, and three new coordination polymers were isolated by a careful selection of the reaction conditions. Coordination polymers {[Zn_3_(DMF)(btrm)(bdc)_3_]·nDMF}_∞_ and {[Zn_3_(btrm)(bdc)_3_]·nDMF}_∞_ containing trinuclear {Zn_3_(bdc)_3_} secondary building units are joined by btrm auxiliary linkers into three-dimensional metal–organic frameworks. The coordination polymer {[Zn(bdc)(btrm)]∙nDMF}_∞_ consists of Zn^2+^ cations joined by bdc^2−^ and btrm linkers into a two-fold interpenetrated network. Upon activation, MOF [Zn_3_(btrm)(bdc)_3_]_∞_ demonstrated CO_2_/N_2_ adsorption selectivity with an ideal adsorbed solution theory (IAST) factor of 21. All three MOF demonstrated photoluminescence with a maximum near 435–440 nm upon excitation at 330 nm.

## 1. Introduction

The separation of carbon dioxide (CO_2_) from dilute gas mixtures, such as flue gas, is considered one of the key problems of sustainable development [1]. Being the product of fossil fuel combustion, CO_2_ is believed to be the major reason for the global climate change and other environmental technogenic alterations. One of the possible ways to prevent uncontrolled CO_2_ release into the atmosphere is its adsorption from the post-combustion flue gas, which usually contains only 15–16% vol. of CO_2_ and nitrogen (73–77% vol.), water vapor (5–7% vol.) and oxygen (3–4% vol.) as other important components [2]. Thus, to achieve economically feasible CO_2_ capture, highly selective adsorbents are required, and currently a lot of effort is being put into the search for metal–organic frameworks with a highly selective adsorption ability toward CO_2_ with high capacity for further storage [3,4,5] or transformation into valuable chemical products [6,7].

Metal–organic frameworks (MOFs) are one of the most perspective classes of coordination polymers due to their luminescent and sensing properties [8,9,10,11,12,13], catalytic activity [14,15,16,17], and high sorption capacity or sorption selectivity toward gases [18,19,20] and liquids [21,22]. Emerging applications of MOFs include targeted drug delivery [23,24], enzyme immobilization [25,26] and bio-imaging [27,28]. The most widespread approach to build MOFs is to use a three-component system of metal ion, di- or polycarboxylate ligand and N,N-bitopic auxiliary ligand that self-assembles into a 3D porous coordination polymer [29,30,31,32]. Bitopic heterocyclic ligands, such as 4,4′-bipyridine [33,34], 4,4′-azo-bis(pyridine) [35,36], are among the most widely used auxiliary ligands. Bis(azol-1-yl)alkanes, such as bis(imidazol-1-yl)alkanes [37,38] and bis(1,2,4-triazol-1-yl)alkanes [39,40,41] are considerably less explored as building blocks for MOFs. Thus, the MOF subset of the Cambridge Structural Database (CSD, more than 114,000 entries) contains only 34 structures where bis(1,2,4-triazol-1-yl)methane (btrm) acts as a linker. Moreover, in only four known coordination polymers, btrm was used in combination with di- or tricarboxylic acids. Zhang et al. prepared 2D sheets assembled from copper(II) ions, ortho-phthalate anions and btrm ligands [42], while Tian et al. reported the formation of a 1D coordination polymer when 1,3,5-benzenetricarboxylate was used as a primary linker [43]. Zinc and cadmium coordination polymers were built using btrm auxiliary ligands and structurally characterized by Tian’s group. A luminescent cadmium–organic framework with CdSO_4_ topology was prepared from isophthalic acid and btrm [44], while a one-dimensional coordination polymer with tube-like chains was formed in the case of zinc ions and 5-sulfonato-benzene-1,3-dicarboxylate linker [45]. In this contribution, we report the results of a detailed investigation of the reaction between zinc nitrate, terephthalic acid (H_2_bdc) and btrm, which allowed us to find the conditions for the selective formation of three new metal–organic frameworks.

## 2. Results and Discussion

### 2.1. Synthesis of the MOFs

The syntheses of MOFs were carried out under solvothermal conditions in dimethylformamide (DMF) at 95–105 °C. The conditions of the experiments were optimized to achieve the crystalline product with the highest yield. In case of polyphase products, the conditions of the experiment were tuned to isolate the pure phases. As a result, three new MOFs were synthesized and characterized by single crystal structure analysis, thermal analysis, luminescence and sorption measurements.

When equimolar amounts of zinc nitrate, btrm and H_2_bdc were heated in a DMF solution (Zn^2+^ concentration [Zn^2+^] in the range of 0.28–0.85 M at 95 °C for 24 h, crystals of the compound {[Zn_3_(btrm)(bdc)_3_(dmf)]∙nDMF}_∞_ (**1**) were obtained. Increasing the heating time to 62 h and slowly cooling the reaction mixture to room temperature allowed us to isolate an additional product–compound {[Zn(btrm)(bdc)]∙3DMF}_∞_ (**3a**) having 1:1:1 Zn:btrm:bdc^2−^ ratio (Figure 1).

Varying the initial concentration of the reagents, we were able to prepare different products. Thus, product **1** formed in the concentration range of 0.28–0.85 M, while at lower zinc concentrations (about 0.03 M) compound {[Zn_3_(btrm)(bdc)_3_]∙DMF}_∞_ (**2**) was obtained. Further experiments have shown that when stored under DMF for 2–3 weeks, compound **2** undergoes a transition into the product {[Zn(btrm)(bdc)]∙3DMF} (**3**) of different composition, which was confirmed by powder XRD analysis (Appendix A).

The purity of all experimental samples was confirmed by comparison of the experimental diffraction patterns with the patterns calculated from single crystal diffraction analysis data (Appendix A).

The thermal analysis of the synthesized compounds revealed that the process of removing the solvent molecules takes place in the range of 70–250 °C (Appendix A). The first stage of thermolysis is associated with the removal of the solvent molecules. The second stage corresponding to degradation of the organic linkers begins immediately after all solvate molecules are lost at about 250 °C. The largest (about 40%) mass loss on the first stage is observed for compound **3**, which is in accordance with its highest free pore volume observed by crystal structure data.

The IR spectra of MOFs **1**–**3** demonstrate the characteristic strong bands near 1500 cm^−1^, corresponding to 1,2,4-triazole ring vibrations and two bands associated with the asymmetric and symmetric carboxylate group vibrations near 1663 (ν_as_COO^−^) and 1601 cm^−1^ (ν_s_COO^−^). The relatively small separation (Δν = 62 cm^−1^) between the last two bands is indicative of the bidentate coordination of the carboxylate groups of the terephthalic acid linkers [46], in accordance with the crystal structures of MOFs **1**–**3**.

### 2.2. Crystal Structures of the MOFs

MOFs **1**, **2**, **3** and **3a** formed single crystals suitable for X-ray structural analysis.

The structures {[Zn_3_(DMF)(btrm)(bdc)_3_]·nDMF}_∞_ (**1**) and {[Zn_3_(btrm)(bdc)_3_]·nDMF}_∞_ (**2**) reveal trinuclear chain-like {Zn_3_(bdc)_3_} units of similar topology. This unit is a relatively common building block of carboxylate MOFs and was previously observed in a similar MOF with 1,3-bis(1,2,4-triazol-1-yl)propane (btrp) linkers [47]. In the case of MOF **1**, central Zn atoms lie on an inversion center, while in MOF **2**, all three Zn atoms are crystallographically independent. In both structures, each {Zn_3_(bdc)_3_} unit is linked to six neighboring ones via bdc^2−^ ligands to form layers (Figure 1a and Figure 2a). These layers are joined into a 3D framework structure via bridging btrm ligands (Figure 1b and Figure 2b). In MOF **1**, two types of {Zn_3_(bdc)_3_} units are observed: the first one is similar to that of MOF **2**, while the second one contains DMF molecules, coordinated to both outer Zn atoms. Thus, the coordination geometry of outer Zn atoms in **1** (of the first {Zn_3_(bdc)_3_} type) and **2** can be considered a tetrahedral consisting of three oxygen and one nitrogen atoms (with average Zn-O/N distance of ca. 2.02 Å) distorted by the inclusion of the fifth oxygen atom with a significantly elongated Zn-O distance (2.54 Å for **1** and 2.43, 2.60 Å for **2**, correspondingly). Outer Zn atoms of the second {Zn_3_(bdc)_3_} type of **1** also have four short Zn-O/N bonds of ca. 2.02 Å, forming a distorted tetrahedral coordination environment, but these atoms additionally coordinate one oxygen atom of bdc^2–^ and one oxygen atom of DMF ligand with elongated Zn-O distances of 2.51 and 2.34 Å, correspondingly. Both frameworks have **hex**/**sqc4** topology [48] (Figure 1c). The structures of MOFs reveal channels filled by DMF molecules. The total free volume in the absence of solvent molecules is estimated to be ca. 45% for **1** and 60% for **2**.

The crystal packing of MOFs {[Zn(bdc)(btrm)]∙2DMF}_∞_ (**3**) and {[Zn(bdc)(btrm)]∙3DMF}_∞_ (**3a**) is very similar. The structure of **3a** can be considered a superstructure of **3** with the doubled unit cell volume, thus the geometries of the frameworks of **3** and **3a** themselves are the same. MOF **3a** reveals two crystallographically independent {Zn(bdc)(btrm)} units of the same geometry; a superstructure manifests itself through different arrangement of solvent DMF molecules. The structures reveal Zn atoms in distorted octahedral environment of two nitrogen atoms of btrm and four oxygen atoms of bdc^2−^ ligands. Zn-O bond lengths differ significantly in the range of 1.98–2.75 Å. Btrm ligands connect Zn atoms to form chains, which are linked by bdc^2−^ ligands into a diamond-like net with hexagon rings (Figure 3a). The **dia**/**sqc6** topology [48] of this net is similar to that in a previously reported four-fold interpenetrated framework {[Zn(bdc)(btrp)]∙1.5DMF}_∞_ [47]. However, in the case of btrm linkers, a two-fold interpenetration of the frameworks is observed in the structures **3** and **3a** (Figure 3b). Thus, despite the shorter methylene chains of btrm compared to btrp, MOFs **3** and **3a** reveal larger voids to be filled by solvent molecules. Three crystallographically independent DMF molecules are located in **3a**, which completely fill channel voids in the structure. In **3**, the solvent molecules are disordered. The free volume of the structures **3** and **3a** in the absence of solvent molecules is estimated to be ca. 55%.

### 2.3. Sorption Properties of the MOFs

To evaluate the permanent porosity and determine the textural properties, the prepared nitrogen adsorption isotherms at 77 K or carbon dioxide adsorption isotherms at 195 K were measured using MOFs **1** and **3** as representative examples. The calculated parameters of the porous structure are given in Table 1. Adsorption–desorption isotherms are shown in Figure 4. The measured nitrogen adsorption isotherm for MOF **1** is Type III (Figure 4a) indicative of the impossibility for nitrogen to penetrate into very narrow channels of these MOFs. Carbon dioxide with a lower kinetic diameter than that of nitrogen can be adsorbed by MOFs **1** (Figure 4a) and **3** (Figure 4b) with the Type I adsorption isotherm; it unambiguously confirms the presence and accessibility for small molecules of narrow channels in MOFs. Carbon dioxide adsorption isotherms for MOFs **1** and **3** are characterized by the significant adsorption–desorption hysteresis of Type H4, which is typical for samples with narrow slit pores.

Since MOF **1** demonstrated a good difference in nitrogen and carbon dioxide adsorption at low temperature, its N_2_/CO_2_ adsorption selectivity was evaluated at ambient temperature. The adsorption isotherms of N_2_ and CO_2_ were measured at 273 K and are shown in Figure 5. As expected, gas uptakes are not high and are 17.9 and 1.6 mL(STP)·g^−1^ for CO_2_ and N_2_, correspondingly (Appendix A). Low uptakes values are a consequence narrow accessible channels and, as a result, low specific surface area. A little adsorption–desorption hysteresis is observed for carbon dioxide isotherm as a consequence of the porous structure with narrow slit pores. There are three commonly used methods to determine the selectivity of adsorption: (i) adsorbed amounts (volumes) ratio; (ii) ratio of Henry constants of adsorption; and (iii) IAST (ideal adsorbed solution theory) calculations, which possess the ability to predict the adsorption selectivity and different total gas pressures and compositions. Henry constants of adsorption were calculated by linearization of the initial parts of isotherms; the values obtained are summarized in Appendix A. For IAST calculations, the isotherms were fitted by Langmuir–Freundlich and Langmuir equations for CO_2_and N_2_, correspondingly (Appendix A). The calculated selectivity factors using all three methods are summarized in Table 2.

According to the IAST calculations, MOF **1** demonstrates moderate CO_2_/N_2_ selectivity with a factor of 21 for an equimolar gas mixture, which further increases up to 40 for N_2_ enriched mixture of 2:8 composition characteristic for flue gas composition (Appendix A). It should also be noted that the adsorption selectivity increases with the total pressure of the equimolar gas mixture (Appendix A). 

### 2.4. Luminescent Properties of the MOFs

The photoluminescence and excitation spectra of MOFs **1**, **2** and **3** in comparison with the free **btrm** ligand in the solid state are shown in Figure 6. Upon excitation at 330 nm (Figure 6b), the MOFs and **btrm** demonstrate blue emission (Appendix A) with similar wide bands associated with the **btrm** intraligand transitions with the maxima near 440 nm (Figure 6a).

Interestingly, in our previous research, the band maxima for btrp and its Zn coordination polymers {[Zn(bdc)(btrp)]·nDMF} and {[Zn_3_(bdc)_3_(btrp)]·nDMF} differ from each other [47], while the positions of the bands for MOFs **1**–**3** are practically identical to the free btrm ligand. This can be due to the higher conformational rigidity of btrm compared to btrp: the geometry of the former differs only slightly [49] from that in complexes **1**–**3**. The angles between the 1,2,4-triazole cycles of btrm units in MOFs **1**–**3** are in a relatively narrow range of 74.0–88.7°, while the geometry of btrp units in the corresponding compounds differs significantly [47]. Differences in the conformations of btrm and crystal packing features influence the quantum yields. The quantum yield increased for MOFs **1** and **2,** compared to the free btrm ligand and decreased for **3**, while for similar compounds with btrp linker, an inverse correlation is observed: the quantum yield increased for diamond-like {Zn} complex and decreased for {Zn_3_} complex (Table 3). 

## 3. Materials and Methods

### 3.1. Synthetic Procudures 

Btrm was synthesized according to the reported method [47]. All reagents were purchased and used without further purification.

#### 3.1.1. Synthesis of Compound {[Zn_3_(btrm)(bdc)_3_DMF]∙nDMF}_∞_ (**1**)

To the mixture of 180.0 mg (1.2 mmol) btrm and 3.0 mL of H_2_bdc (0.4 M) DMF solution (1.2 mmol) in a glass vial, 1.2 mL of Zn(NO_3_)_2_·6H_2_O (1.0 M) DMF solution (1.2 mmol) was added. The mixture was stirred for several minutes at room temperature until the complete dissolution of all reagents. The vial was placed in the oven at 95 °C for 24 h. After this time, the vial was removed from the oven and cooled to room temperature. Colorless prismatic crystals formed on the bottom. The crystals were washed twice with 20 mL of DMF and then stored in the glass vial filled with a few milliliters of DMF. The yield was about 220 mg (16%). IR bands, cm^−1^: 3121, 2928, 1663, 1601, 1505, 1385 (broad), 1281, 1130, 1100, 1019, 999, 887, 826, 749, 675 (broad). Elemental analysis: found, %: C 43.1, H 4.4, N 12.4; and calculated ([Zn_3_(btrm)(bdc)_3_DMF]∙nDMF, n = 3), %: C 43.9, H 4.1, N 12.4.

Reproducing the reaction at the same conditions for a longer period of time (62 h) led to the formation of MOF **1** as a major product and a few crystals of MOF [Zn(btrm)(bdc)]·nDMF (**3a**) as a minor product.

#### 3.1.2. Synthesis of Compound {[Zn_3_(btrm)(bdc)_3_]∙nDMF}_∞_ (**2**)

The solution of 1.6 mL H_2_bdc (0.4M) in DMF (0.64 mmol) was added to the 96.0 mg of btrm ligand (0.64 mmol) in a glass vial and stirred for a few minutes. Then 16 mL of Zn(NO_3_)_2_·6H_2_O (0.04 M) solution in DMF (0.64 mmol) was added and stirred for a few hours at room temperature. The solution was placed in an oven at 95 °C for 48 h. The crystals formed were washed twice with 20 mL of DMF. The yield was 60 mg (10%). IR bands, cm^−1^: 3123, 2928, 1958, 1669, 1603, 1530, 1499, 1441, 1387, 1348, 1279, 1210, 1132, 1090, 997, 966, 889, 828, 750, 677, 657. Elemental analysis: found, %: C 42.6, H 4.3, N 10.6; and calculated ([Zn_3_(btrm)(bdc)_3_]∙nDMF, n = 1), %: C 42.2, H 2.8, N 10.8.

#### 3.1.3. Synthesis of Compound {[Zn(btrm)(bdc)]·nDMF}_∞_ (**3**)

The procedure of synthesis (including loadings and concentrations of solutions used) was similar to the synthesis of compound **2**, but the time of heating before filtration was reduced to 24 h. The small number of crystals of **2** were filtered off and washed twice with 10 mL of DMF, and the resulting filtrate solution (about 40 mL) was left at room temperature. After a few weeks, small colorless crystals were formed on the bottom and walls of the vial. The yield was 230 mg (60%). IR bands, cm^−1^: 3121, 2928, 1663, 1601, 1505, 1385(broad), 1281, 1130, 1100, 1019, 999, 887, 826, 749, 675 (broad). Elemental analysis: found, %: C 44.1, H 5.1, N 20.7; and calculated ([Zn(btrm)(bdc)]∙nDMF, n = 3), %: C 44.1, H 5.2, N 21.0.

### 3.2. Methods of Characterization

Elemental analyses were carried out on Eurovector EuroEA 3000 analyzer (Elementar Analysensysteme GmbH, Langenselbold, Germany). Infrared (IR) spectra of solid samples as KBr pellets were recorded on a FT-801 spectrometer (4000–550 cm^−1^_,_ Kailas OU, Tallin, Estonia). Polycrystalline samples were studied in 2θ range 5–60° on a DRON RM4 powder diffractometer (Burevestnik, Saint Petersburg, Russia) equipped with a CuKα source (λ = 1.5418 Å) and graphite monochromator for the diffracted beam. Thermogravimetric measurements were carried out on a NETZSCH thermobalance TG 209 F1 Iris (Erich NETZSCH GmbH & Co. Holding KG, Selb, Germany). Open Al_2_O_3_ crucibles were used (loads 10–20 mg, He atmosphere, heating rate 10 K·min^−1^). Room temperature excitation and emission spectra were recorded with a Horiba Jobin Yvon Fluorolog 3 (HORIBA Jobin Yvon SAS, Edison, NJ, USA) photoluminescence spectrometer equipped with 450W ozone-free Xe-lamp (HORIBA Jobin Yvon SAS, Edison, NJ, USA), cooled PC177CE-010 photon detection module (HORIBA Jobin Yvon SAS, Edison, NJ, USA) with a PMT R2658 and double grating excitation and emission monochromators (HORIBA Jobin Yvon SAS, Edison, NJ, USA). Quantum yields were determined using Quanta-φ integrating sphere (HORIBA Jobin Yvon SAS, Edison, NJ, USA). Excitation and emission spectra were corrected for source intensity (lamp and grating) and emission spectral response (detector and grating) by standard correction curves. 

### 3.3. X-ray Structure Determination

Single-crystal XRD data for the complexes were collected by a Bruker Apex DUO diffractometer (Bruker Corporation, Billerica, MA, USA) equipped with a 4K CCD area detector using the graphite-monochromated MoKα radiation (λ = 0.71073 Å) (Table 1). The φ- and ω-scan techniques were employed to measure intensities. Absorption corrections were applied with the use of the SADABS program [50]. The crystal structures were solved by direct methods and refined by full-matrix least squares techniques with the use of the SHELXTL package [51] and Olex2 GUI [52]. Atomic thermal displacement parameters for non-disordered non-hydrogen atoms were refined anisotropically. The positions of hydrogen atoms were calculated corresponding to their geometrical conditions and refined using the riding model. DFIX, DANG, FLAT and RIGU restrains and EADP constrains were applied to atoms of disordered units where needed. In **2** and **3**, some solvent DMF molecules appeared to be highly disordered, and it was difficult to model their positions reliably. Therefore, the structures were treated via the Solvent Mask procedure [53] to remove the contribution of the electron density in the solvent regions from the intensity data. For **1**, the potential solvent accessible void volume was estimated to be 279 Å^3^ and the electron count per unit cell was 70, which were assigned to 2 molecules per unit cell and one DMF molecule per formula unit. For **2**, the corresponding void volume was 7216 Å^3^ with the electron count of 2571, which were assigned to 64 molecules per unit cell and 8 DMF molecules per formula unit. For **3**, the potential solvent accessible void volume was estimated to be 3349 Å^3^ and the electron count per unit cell was 866, which were assigned to 24 molecules per unit cell and 3 DMF molecules per formula unit. The crystallographic parameters and crystal data collection and structure refinement data are summarized in Appendix A. The topology of the frameworks was analyzed using the TopCryst system [54].

### 3.4. Study of Gas Sorption Properties

#### 3.4.1. Methods for the Experimental Study of Adsorption-Desorption Isotherms

Analysis of surface area and porous structure of MOFs was performed by nitrogen adsorption at 77 K or carbon dioxide adsorption at 195 K technique using Autosorb iQ instrument (Quantochrome, Boynton Beach, FL, USA), equipped with the cryostat CryoCooler (Cryomech, Syracuse, NY, USA) were used to adjust the temperature with ±0.05 K accuracy. 

Before gas sorption experiments, samples were activated in a dynamic vacuum (10^−8^ bar) at 30 °C during 1 h. Nitrogen or carbon dioxide adsorption–desorption isotherms were measured within the range of relative pressures of 10^−6^ to 0.995. The specific surface area was calculated from the data obtained based on the conventional BET and Langmuir models.

Gas adsorption isotherm measurements at 273 K were carried out volumetrically on Quantochrome’s Autosorb iQ equipped with thermostat TERMEX Cryo VT-12 (TERMEX Ltd., Tomsk, Russia) to adjust the temperature with ±0.1 K accuracy. Adsorption−desorption isotherms were measured within the range of pressures from 1 to 800 torr. The database of the National Institute of Standards and Technology [55] was used as a source of p-V-T relations at experimental pressures and temperatures.

#### 3.4.2. Evaluation of the Adsorption Selectivity

Adsorption selectivities for CO_2_/N_2_ binary gas mixture were calculated using three different approaches:As the molar ratio of the adsorption quantities at the relevant partial pressures of the gases,
$$S=\frac{n_{1}/n_{2}}{p_{1}/p_{2}},$$
where *S* is the selectivity factor, *n_i_* represents the adsorbed amount of component *i*, and *p_i_* represents the partial pressure of component *i*.

2.As a ratio of Henry constants which corresponds to the slope of the adsorption isotherm at very low partial pressures,


$$S=\frac{K_{H1}}{K_{H2}}$$


3.By an ideal adsorbed solution theory (IAST). The relationship between *P*, *y_i_* and *x_i_* (*P*—the total pressure of the gas phase, *y_i_*—mole fraction of the *i* component in the gas phase, *x_i_*—mole fraction of the *i* component in the absorbed state) is defined according to the IAST theory [56]:


$$\int_{p=0}^{p=\frac{Py_{1}}{x_{1}}}n_{1}\left(p\right)dlnp=\int_{p=0}^{p=\frac{Py_{2}}{x_{2}}}n_{2}\left(p\right)dlnp$$


In this case, the selectivity factors were determined as
$$S=\frac{y_{2}x_{2}}{y_{1}x_{1}}=\frac{x_{1}\left(1-y_{1}\right)}{y_{1}\left(1-x_{1}\right)}$$

For IAST calculations, the adsorption isotherms were primarily fitted by an appropriate model. The models used and the mathematical equations for each model are given in Appendix A.

## 4. Conclusions

In summary, the careful control of the reaction conditions in a three-component system terephthalic acid-bis(1,2,4-triazol-1-yl)methane-zinc nitate in DMF allowed to prepare and structurally characterize three new metal-organic frameworks, which extend the so far underrepresented family of bis(1,2,4-triazol-1-yl)methane-based coordination polymers. The gas adsorption properties of btrm-based MOFs were studied for the first time, and MOF {[Zn_3_(btrm)(bdc)_3_DMF]∙nDMF}_∞_ demonstrated selective adsorption of carbon dioxide over nitrogen with an IAST selectivity factor of 21. Despite the moderate adsorption capacity preventing it from being applicable for the real industrial separation of flue gas, the results obtained are interesting from a fundamental point of view and demonstrate the potential of MOFs based on bis(1,2,4-triazol-1-yl)methane ligand for selective gas adsorption. Thus, further efforts should be made in looking for compounds with less interpenetrated structures and wider channels. In addition, the MOFs demonstrated blue photoluminescence with moderate quantum yields up to 12 %, indicating the potential of bis(1,2,4-triazol-1-yl)alkane ligands in combination with aromatic dicarboxylate ligands for the optimization of the photophysical properties and the preparation of efficient luminophors.

## Data Availability

Experimental data associated with this research are available from the authors. Crystallographic data for the structural analysis have been deposited with the Cambridge Crystallographic Data Centre, CCDC No. 2207811 for compound **1,** 2207814 for compound **2,** 2207813 for compound **3** and 2207812 for compound **3a**. Copies of the data can be obtained free of charge from the Cambridge Crystallographic Data Centre, 12 Union Road, Cambridge CB2 1EZ, UK (Fax: +44-1223-336-033; e-mail: deposit@ccdc.cam.ac.uk).

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
