# Peer review of "Structural Diversity and Carbon Dioxide Sorption Selectivity of Zinc(II) Metal-Organic Frameworks Based on Bis(1,2,4-triazol-1-yl)methane and Terephthalic Acid"

_molecules, 2022, doi:10.3390/molecules27196481_

Round 1

Reviewer 1 Report

The authors synthesized a series of bis(1,2,4-triazol-1-yl)methane-based coordination polymers and characterized them in the aspects of luminescence, gas adsorption, gas separation, et al. This work is technically sound and I therefore recommend its publication in Molecules after some revisions: 

1.     Please supplement the meaning of CO2/N2 gas separation and related research progress in Introduction part.

2.     Please specify the potential applications of the reported luminescent materials.

3.     The full name of IAST should be given in the Abstract when it first appeared.

4.     Please keep all the figures with uniform font style and size. For example, the font was totally different between Fig4a and Fig4b; The format of font in Fig5 was different with other figures.

5.     Please modify the symbol of structure parameters, such as "a,b,c", into italic.

Author Response

  1. Please supplement the meaning of CO2/N2 gas separation and related research progress in Introduction part.

The introduction part was extended its first part is now devoted the problem of CO2 capture and separation.

  1. Please specify the potential applications of the reported luminescent materials.

The conclusions section was extended by a note on the potential of the prepared MOFs as the starting points for further optimization of the photophysical properties in search of the efficient luminophores.

  1. The full name of IAST should be given in the Abstract when it first appeared.

Corrected, full name is given.

  1. Please keep all the figures with uniform font style and size. For example, the font was totally different between Fig4a and Fig4b; The format of font in Fig5 was different with other figures.

Formatting and font in the figures was changed and made uniform.

  1. Please modify the symbol of structure parameters, such as "a,b,c", into italic.

Corrected.

Reviewer 2 Report

1. Please unify the full concept on coordination polymer, metal-organic framework etc.

2. Please also give the topology analysis for the coordination polymer 1 and 2.

3. Please compared the similar MOF on the BET.

4. Some MOF on the gas adsorption could be cited and compare, such as Inorg. Chem. 2021, 60, 18593-18597ï¼› Cryst. Growth Des. 2022, 22, 4018-4024ï¼› Inorg. Chem. 2022, 61, 13234-13238ï¼›Chem. Sci., 2022, 13, 5130-5140

5. Remove the Table 4 from the main text into ESI.

6. Please provide TAG and IR and analyze them.

7. Please illustrate the excited emission for all the PL.

Author Response

  1. Please unify the full concept on coordination polymer, metal-organic framework etc.

The term MOF is now used throughout the manuscript.

  1. Please also give the topology analysis for the coordination polymer 1 and 2.

The topologies of the frameworks were added to the text and an additional figure (Fig. 1c) was added.

  1. Please compared the similar MOF on the BET.

None of the papers devoted to MOFs based on btrm ligands report BET surface areas, thus our work is the first example of gas sorption properties of btrm-based MOFs. A note on this was added to the conclusions section.

  1. Some MOF on the gas adsorption could be cited and compare, such as Inorg. Chem. 2021, 60, 18593-18597ï¼› Cryst. Growth Des. 2022, 22, 4018-4024ï¼› Inorg. Chem. 2022, 61, 13234-13238ï¼›Chem. Sci., 2022, 13, 5130-5140

These relevant references were cited as Refs. 11-13 and 17.

  1. Remove the Table 4 from the main text into ESI.

This table was moved to the Supplementary materials.

  1. Please provide TAG and IR and analyze them.

The TGA analysis is given on page 2 (and Figs. S5-S6). Coordinated and guest DMF molecules are lost simultaneously, thus it is not possible to quantitatively distinguish these two steps and give a more detailed analysis. The IR spectra analysis was added in lines 82-87 and it discusses the positions of characteristic btrm and bdc2- bands.

  1. Please illustrate the excited emission for all the PL.

The images are included in the supplementary materials as Figure S9.

Round 2

Reviewer 2 Report

the authors have addressed all the comment.